# Unique Features of the Immune Response in BTBR Mice

**DOI:** 10.3390/ijms232415577

**Published:** 2022-12-08

**Authors:** Anastasia Mutovina, Kseniya Ayriyants, Eva Mezhlumyan, Yulia Ryabushkina, Ekaterina Litvinova, Natalia Bondar, Julia Khantakova, Vasiliy Reshetnikov

**Affiliations:** 1Institute of Cytology and Genetics (ICG), Siberian Branch of Russian Academy of Sciences (SB RAS), Prospekt Akad. Lavrentyeva 10, 630090 Novosibirsk, Russia; 2Physical Engineering Faculty, Novosibirsk State Technical University, Prospekt Karl Marx, 20, 630073 Novosibirsk, Russia; 3Department of Natural Sciences, Novosibirsk State University, Pirogova Street 2, 630090 Novosibirsk, Russia; 4Department of Biotechnology, Sirius University of Science and Technology, 1 Olympic Avenue, 354340 Sochi, Russia

**Keywords:** Poly I:C, LPS, neuroinflammation, BTBR

## Abstract

Inflammation plays a considerable role in the pathogenesis of many diseases, including neurodegenerative and psychiatric ones. Elucidation of the specific features of an immune response in various model organisms, and studying the relation of these features with the behavioral phenotype, can improve the understanding of the molecular mechanisms of many psychopathologies. In this work, we focused on BTBR mice, which have a pronounced autism-like behavioral phenotype, elevated levels of oxidative-stress markers, an abnormal immune response, several structural aberrations in the brain, and other unique traits. Although some studies have already shown an abnormal immune response in BTBR mice, the existing literature data are still fragmentary. Here, we used inflammation induced by low-dose lipopolysaccharide, polyinosinic:polycytidylic acid, or their combinations, in mice of strains BTBR T+Itpr3tf/J (BTBR) and C57BL6/J. Peripheral inflammation was assessed by means of a complete blood count, lymphocyte immunophenotyping, and expression levels of cytokines in the spleen. Neuroinflammation was evaluated in the hypothalamus and prefrontal cortex by analysis of mRNA levels of proinflammatory cytokines (tumor necrosis factor, *Tnf*), (interleukin-1 beta, *Il-1β*), and (interleukin-6, *Il-6*) and of markers of microglia activation (allograft inflammatory factor 1, *Aif1*) and astroglia activation (glial fibrillary acidic protein, *Gfap*). We found that in both strains of mice, the most severe inflammatory response was caused by the administration of polyinosinic:polycytidylic acid, whereas the combined administration of the two toll-like receptor (TLR) agonists did not enhance this response. Nonetheless, BTBR mice showed a more pronounced response to low-dose lipopolysaccharide, an altered lymphocytosis ratio due to an increase in the number of CD4+ lymphocytes, and high expression of markers of activated microglia (*Aif1*) and astroglia (*Gfap*) in various brain regions as compared to C57BL6/J mice. Thus, in addition to research into mechanisms of autism-like behavior, BTBR mice can be used as a model of TLR3/TLR4-induced neuroinflammation and a unique model for finding and evaluating the effectiveness of various TLR antagonists aimed at reducing neuroinflammation.

## 1. Introduction

For a long time, it has been believed that the role of the immune system is limited to the formation of protective immunity against various infections, inflammatory reactions, or cancer. Nevertheless, subsequent studies have shown a close relation between the nervous and immune systems [1]. Biomolecules that are traditionally associated with immunological function are expressed in the nervous system and vice versa [2,3,4,5]. In various neuropsychiatric disorders, e.g., in brain tissues, signs of persistent local neuroinflammation have been found [6,7], which can impair the normal development of the nervous system, the formation of synaptic connections and, ultimately, the behavior of the individual [8,9].

Toll-like receptors (TLRs) are the best known innate-immune-system receptors that recognize pathogen-associated molecular patterns [10]. When a TLR is stimulated by microbial products, innate-immune-response cells, such as resident macrophages, dendritic cells, mast cells, eosinophils, and neutrophils, are activated first. This leads to an increase in the synthesis of proinflammatory mediators, such as tumor necrosis factor (TNF), interleukin-6 (IL-6), and reactive oxygen species (ROS). In some cases, the innate immune response is unable to eliminate the infection and requires the induction of an adaptive immune response. Elevated levels of proinflammatory cytokines (TNF, IL-1, IL-6, IL-8, and IL-12), chemokines, and nitric oxide (NO) activate functions of antigen-presenting cells and enable the induction of an adaptive immune response, in which both T and B lymphocytes play a decisive role. Due to widespread expression of TLRs in the brain by various cell types, including the brain’s local sentinel innate-immunity cells such as microglia and astrocytes [10], it is becoming clear that dysregulation of a TLR response can cause neurodevelopmental disorders [11]. Activation of neuronal TLR3, TLR7, and TLR8 by nucleic acids has been reported to affect neuronal connectivity and to alter synapse formation and brain function, thereby inducing deficits related to neuropsychiatric disorders [11,12,13,14,15,16,17]. Activation of TLR4 pathways may cause chronic inflammation and overproduction of ROS and reactive nitrogen species (RNS) and oxidative and nitrosative stress [18,19].

In the present study, we examined differences in the acute immune response to a single intraperitoneal injection of a TLR3 agonist (polyinosinic:polycytidylic acid [polyI:C]) and/or a TLR4 agonist (lipopolysaccharide; LPS) between BTBR and C57BL6/J (B6) mice. The BTBR mouse strain is one of the most valid models of autism spectrum disorders (ASDs) because these mice exhibit the most prevalent ASD characteristics, such as deficiencies in social behaviors and reduced or unusual ultrasonic vocalizations as well as greater repetitive self-grooming [20,21]. Additionally, they demonstrate aberrant immunological responses, which are all diagnostic of an ASD [22,23]. BTBR mice exhibit altered proinflammatory-mediator expression, which is linked to the severity of behavioral problems [24]. These include higher levels of serum immunoglobulin G (IgG) and anti-brain antibodies; elevated expression of cytokines, especially IL-6, IL-17, IL-1β, and TNF; and an increased percentage of major histocompatibility complex (MHC) class II–expressing microglia as compared to B6 mice [25]. Moreover, research has shown that BTBR mice have a unique immunological profile along with an altered T helper profile [26]. Notably, bone marrow transplantation from social C57BL/6 mice to BTBR mice leads to the normalization of their behavior [27], implying a connection between the behavioral and immune deficiencies. C57BL6/J mice are a highly social strain of mice and are frequently used as a control in experiments on BTBR mice [28].

We injected LPS and polyI:C systemically into BTBR and B6 mice to set up a model of acute inflammation in the periphery and in the brain (Figure 1). A complete blood count with differential lymphocyte immunophenotyping and regional expression levels of cytokines and neural, microglial and astrocyte activation molecules in the prefrontal cortex (PFC) and hypothalamus of LPS- and polyI:C-treated mice is presented here. The choice of these brain regions is explained by the fact that they are involved in the regulation of the stress response, and hypothalamic and cortical inflammation may also be associated with behavioral disturbances and psychiatric disorders [29,30,31]. In addition, the hypothalamus plays a crucial role in systemic homeostatic regulation, whereas inflammation may disrupt homeostatic regulation [32,33].

## 2. Results

### Features of a Peripheral Immune Response in BTBR Mice

After the administration of the proinflammatory agents, both strains of mice developed sickness behavior within 16 h, accompanied by diarrhea and apathy.

The development of peripheral inflammation was assessed by means of changes in parameters of the complete blood count, in lymphocyte populations, and in the expression of proinflammatory cytokines in the spleen at 2 and 16 h after the administration of each mimetic. Overall, despite comparable parameters of the complete blood count between strains BTBR and C57BL6/J (further in the text B6), there were differences in the response to inflammatory stress (Figure 2).

The counts of leukocytes, lymphocytes, monocytes, and granulocytes reacted to the injection of the mimetics in a strain-specific and TLR-specific manner. For instance, following the administration of the TLR4-specific mimetic (LPS), already after 2 h, BTBR mice experienced short-term leukopenia, lymphopenia, and slight monocytopenia, characteristic of the acute phase of inflammation, which disappeared within 16 h after the administration of the mimetic. By contrast, in B6 mice, a decrease in these counts was observed only after 16 h. The response of granulocytes to the injection of LPS did not differ between the two mouse strains. With the TLR3-specific stimulation (by polyI:C), a different picture was observed: in BTBR mice, after 2 h, there was a decrease in the counts of leukocytes and lymphocytes, and these low numbers persisted for 16 h. The numbers of monocytes and granulocytes did not change. In B6 mice, after 2 h, along with leukopenia and lymphopenia, the number of granulocytes decreased too. These alterations in B6 mice persisted for 16 h, and there was also a trend toward a decrease in the monocyte count. These data indicated a more prolonged inflammatory response to TLR3-specific stimulation than to the TLR4-specific stimulation; this reaction was more pronounced in the B6 strain.

The combined injection of LPS and polyI:C into BTBR mice did not have a significant effect on the cellular composition of peripheral blood when compared with each mimetic, with the exception of an increase in the granulocyte count after 16 h in BTBR mice. In B6 mice, the combination of LPS and polyI:C, when compared with each mimetic alone, did not significantly affect cell responses after 2 and 16 h.

The injection of each proinflammatory agent alone or of their combination led to a sharp decrease in the number of platelets after 2 h in the B6 strain but not in BTBR mice. Nonetheless, no differences were detectable between the two types of TLR stimulation. The detected decrease disappeared at 16 h after administration (a return to control values). Such a sharp drop of the platelet count during inflammation is usually associated with increased clotting intended to limit the volume of the affected tissue. It is possible that the hyperergic reaction of platelets contributed to the slowness of the response of B6 mice’s blood cells to the inflammatory agents.

Thus, our results indicate interstrain differences in the response to TLR-specific stimulation. BTBR mice showed significantly greater sensitivity to TLR4-specific stimulation even at low doses of the mimetic, but the reactions were short-lived and mostly disappeared after 16 h. Exposure to the TLR3 mimetic altered cellular composition of peripheral blood and the change was more prolonged, being more pronounced in B6 mice. The combined injection of polyI:C and LPS did not enhance the cellular responses (seen with each mimetic alone) in both strains, indicating the absence of an additive effect of the two mimetics.

Next, we assessed changes in the sizes of major lymphocyte subpopulations after administration of the TLR mimetics (Figure 3). In a steady state, BTBR mice had more CD45^+^ leukocytes in their peripheral blood than B6 mice did. Additionally, in BTBR mice, a decrease in the number of CD45^+^ cells was observed after 16 h, only when exposed to the combination of the mimetics. By contrast, in B6 mice, even individual administration of any of the mimetics diminished the count of CD45^+^ leukocytes, and the effect was more pronounced in the polyI:C group.

In addition, differences were noted in baseline counts of lymphocytes: in BTBR mice, CD3^+^ T-lymphocytes predominate, whereas in B6 mice, CD19^+^ B-lymphocytes are dominant, which may indicate a predisposition to a cellular or humoral type of immune response, respectively. In BTBR mice, administration of LPS, polyI:C, or their combination resulted in significant comparable increases in the number of CD3^+^ cells, and a decrease in the number of CD19^+^ lymphocytes at 2 h after the injection (Table 1). On the other hand, after 16 h, all parameters returned to control values, with the exception of the number of CD19^+^ lymphocytes after the administration of polyI:C. In B6 mice, no effects of drug administration on the number of CD3^+^ cells were found. Nonetheless, a significant decline of the number of B lymphocytes was noted at 16 h after the injection in groups polyI:C and polyI:C+LPS. Consequently, the findings confirm (i) the sensitivity of strains BTBR and B6 to stimulation of the T-cell or B-cell type of immune response, respectively; (ii) a more prolonged effect of TLR3 stimulation on the immune response; and (iii) the sensitivity of peripheral immunocompetent cells in the BTBR strain to TLR4 stimulation, all without an additive effect of the mimetics.

It was also found that BTBR mice in a steady state have a reduced count of CD8^+^ cytotoxic T lymphocytes and an elevated count of CD4^+^ T helper cells as well as a lower CD8^+^/CD4^+^ ratio as compared to B6 mice. Of note, despite the previously described sensitivity of BTBR mice to TLR4 stimulation, the numbers of CD4^+^ and CD8^+^ T lymphocytes were not altered at both time points after the administration of LPS. The administration of polyI:C or of the combination polyI:C+LPS led to a decline of the number of CD8^+^ cells and an increase in the CD4^+^ cell count as compared to the control in both strains of mice at 16 h after the injection; this phenomenon is typical for the acute phase of inflammation and was more pronounced in BTBR mice. CD4^+^ T cells mediate adaptive immunity against a variety of pathogens and determine the type of immune response. Depending on the type of T helper cells, they provide help to phagocytes (type 1), to B cells, eosinophils, and mast cells (type 2), or to nonimmune tissue cells, including stromal and epithelial cells (type 3) [34]. If not adequately regulated, CD4^+^ T cells can be also involved in autoimmunity, asthma, and other allergic responses.

Thus, the flow cytometry data show that low-dose LPS has a short-term effect on lymphocytes only in BTBR mice. The effects of the TLR3 mimetic were more pronounced and prolonged than those of TLR4 activation. As for the sensitivity of the mouse strains, similarly to the findings about the complete blood count, the B6 strain showed higher sensitivity of TLR3. Combining polyI:C and LPS did not enhance the response of lymphocytes in both strains, which further confirms the absence of an additive effect of the two mimetics.

Alterations of expression of genes of proinflammatory cytokines *Tnf, Il-1β,* and *Il-6* and neuronal early response gene (*cFos)* after the induced acute inflammation were also examined by qRT-PCR in the spleen (Figure 4). Analysis of proinflammatory cytokines’ gene expression in the spleen uncovered significant effects of an interaction of factors “strain” and “group” on *Il-6* at 2 h after the induction of inflammation (F(3;32) = 29.48, *p* < 0.001). There was a significant effect of the “strain” on *Tnf* at 2 and 16 h (F(1;31) = 14.84, *p* < 0.001 and F(1;45) = 5.82, *p* < 0.05), on IL-1β at 16 h (F(1;43) = 7.12, *p* < 0.05), and on *Il-6* at 2 h (F(1;32) = 48.94, *p* < 0.001).

The *cFos* mRNA level significantly changed after administration of LPS, polyI:C, or their combination at 2 h postinjection in BTBR mice, whereas in B6 mice, only polyI:C significantly raised *cFos* expression at 2 h; at 16 h, B6 mice showed a significant increase after the combination of LPS and polyI:C was injected, whereas BTBR mice demonstrated upregulation of *cFos* after the injection of only polyI:C. *Tnf* mRNA baseline levels were significantly higher in B6 mice and increased after all kinds of induced inflammation at 2 h after the injection. Expression of proinflammatory cytokines *Tnf, Il-1β*, and *Il-6* increased similarly after LPS, polyI:C, and their combination in BTBR mice at 2 h postinjection, and after 16 h these parameters returned to normal; only *Tnf* still showed significant upregulation after polyI:C. By contrast, in B6 mice, *Il-1β* and *Il-6* mRNA levels went up only after the injection of polyI:C and of the combination of LPS and polyI:C at 2 h; at 16 h, only *Il-1β* still showed significant overexpression after the combined injection. Notably, the production of *Tnf* at 2 h after exposure to any mimetic or their combination did not differ between the two strains. The BTBR strain was more prone to the production of *Il-6*, whereas B6 mice to the production of *Il-1b* in the first 2 h after exposure to polyI:C or its combination with LPS. In this context, in BTBR mice, the combination polyI:C+LPS significantly increased the expression of *Il-6*.

Thus, in the analysis of indicators of peripheral inflammation, we demonstrated that during TLR4 activation, there are changes only in the BTBR strain, but they are transient. Both strains proved to be sensitive to the effects of the TLR3 mimetic; however, more pronounced changes were seen in strain B6.

The analysis of the proinflammatory genes revealed significant effects of an interaction of factors “strain” and “group” on *cFos* in the hypothalamus at 2 h (F(3;32) = 5.29, *p* < 0.01), on *Aif1* in the hypothalamus at 16 h (F(3;46) = 5.26, *p* < 0.01), on *Aif1* in the PFC at 2 h (F(3;30) = 5.11, *p* < 0.01), and on *Il-6* in the hypothalamus and PFC at 16 h (F(3;41) = 4.92, *p* < 0.01, F(3,43) = 4.59, *p* < 0.01). Significant differences between strains were observed in mRNA levels of *cFos* in the PFC at 2 h (F(1;31) = 11.28, *p* < 0.01) and 16 h (F(1;45) = 9.98, *p* < 0.01), *Aif1* in the hypothalamus at 2 h (F(1;32) = 304.64, *p* < 0.001) and 16 h (F(1;46) = 48.39, *p* < 0.001), in the PFC at 2 h (F(1;30) = 79.82, *p* < 0.001) and 16 h (F(1;46) = 22.95, *p* < 0.001), *Gfap* in the hypothalamus at 2 h (F(1;32) = 17.7, *p* < 0.001) and 16 h (F(1;46) = 19.48, *p* < 0.001) and in the PFC at 2 h (F(1;31) = 38.27, *p* < 0.001), *Il-6* in the hypothalamus at 16 h (F(1;41) = 72.97, *p* < 0.001) and in the PFC at 2 h (F(1;30) = 11.56, *p* < 0.01) and 16 h (F(1;43) = 15.72, *p* < 0.001), and *IL-1β* in the hypothalamus at 2 h (F(1;32) = 8.63, *p* < 0.01).

Levels of *cFos* mRNA rose significantly after the administration of polyI:C and its combination with LPS at 2 h post-injection in the hypothalamus of both B6 and BTBR mice (Figure 5). In the PFC, B6 mice showed a similar increase in *cFos* mRNA, albeit at a lower level, whereas in BTBR mice, only polyI:C injection led to significant upregulation. At 16 h, no significant difference was found.

*Aif1* baseline expression differed between BTBR and B6 mice: it was higher in the hypothalamus and lower in the PFC of BTBR mice. Changes in mRNA levels emerged after 16 h in the hypothalamus of BTBR mice and in the PFC of both strains after the polyI:C or combined injection. The induced systemic inflammation caused more extensive activation of microglia in BTBR mice than in B6 mice.

Hypothalamic levels of *Gfap* were not affected by the induced inflammation, although its baseline level was much higher in BTBR mice. In the PFC, the *Gfap* mRNA level increased in BTBR mice at 2 h after the administration of polyI:C or combined injection. This upregulation continued up to the time point 16 h, whereas in B6 mice, the *Gfap* level rose only at 16 h after all types of injection. Therefore, astrocytes in the PFC react to the induced inflammation earlier in BTBR mice than in B6 mice.

In BTBR mice, astrocytes reacted much earlier than microglia did, whereas in B6 mice, these cell types appear to undergo more or less simultaneous activation.

In contrast to peripheral tissues, in the brain, LPS did not cause any changes of mRNA levels in brain cells (namely, in activation markers or cytokines) in BTBR mice. On the contrary, in B6 mice, LPS effects were noticeable at 16 h postinjection: later than polyI:C effects. Nevertheless, even in this case, the effect of LPS was more pronounced relative to polyI:C. polyI:C and its combination with LPS induced overexpression of proinflammatory cytokines *Tnf, Il-1β*, and *Il-6* at 2 h in the hypothalamus and PFC of both strains. Besides, high levels of *TNF* mRNA persisted for up to 16 h after the combined injection in BTBR mice and after the injection of polyI:C and combination in the PFC of B6 mice. Moreover, LPS caused mRNA overexpression of *Tnf* and *Il-1β* at 16 h in the hypothalamus and PFC of B6 mice only. High *Il-6* mRNA levels persisted for up to 16 h in BTBR mice only.

## 3. Discussion

Studying strain-specific responses to TLR agonists that mimic bacterial or viral pathogens in mice may improve our understanding of individual immune responses to bacteria, viruses, or vaccination. In our work, we found that BTBR mice show a more pronounced response to a low dose of LPS as compared to B6 mice (i), which manifests itself in an increase in the number of T lymphocytes, a decrease in the number of B lymphocytes, an overall decline of the counts of leukocytes and lymphocytes and overexpression of proinflammatory cytokines in the spleen as early as 2 h after the administration of LPS. Second, we detected marked interstrain differences in lymphocyte populations between the two strains of mice in a steady state (at baseline) (ii). BTBR mice were found to have fewer B lymphocytes and more T lymphocytes, owing to a greater number of CD4^+^ lymphocytes, while the number of CD8^+^ lymphocytes, on the contrary, is reduced in comparison with B6 mice (iii). Additionally, quantification of gene expression in the brain revealed that BTBR mice show increased expression of markers of activated microglia (*Aif1*) and astroglia (*Gfap*) in the hypothalamus in a steady state and increased reactivity of the expression of *Aif1* in the hypothalamus and of *Gfap* in the PFC at 2 h after exposure to polyI:C or the combination of polyI:C and LPS (iv). Cytokine expression in the hypothalamus and PFC in response to the mimetic injection also differed between the two strains in a region-dependent manner (v). Taken together, our results improve the understanding of (1) specific features of the immune system of BTBR mice in a steady state and (2) dynamic alterations of immunoreactivity in response to bacterial and viral mimetics.

Our results indicate that in BTBR mice, after the administration of polyI:C or of its combination with low-dose LPS, most of the effects on the peripheral immune system and on the expression of *c-Fos*, *Il-1β*, *Il-6*, and *Tnf* are detectable in the hypothalamus, PFC, and spleen at 2 h after the mimetic injection. The expression of these genes at 16 h after the injection in most cases returned to baseline. On the other hand, the expression of the *Aif1* gene (a marker of microglial activation) was observed in most cases only at 16 h after the administration of the mimetics. Remarkably, despite the observed effects on immunocompetent cells and on the expression of proinflammatory genes in the spleen, the administration of LPS in BTBR mice did not affect neuroinflammation. These results are consistent with other studies, which show that the highest activation of cytokine production is seen at 1.5–3.0 h after induction by inflammatory agents [35,36]; furthermore, the peak of glia activation and changes in the profile of peripheral blood lymphocytes are observed after 6–24 h [36,37].

In our work, we noticed that the administration of a low dose of LPS in combination with polyI:C had almost no influence on the proinflammatory cytokine profile of both strains of mice as compared with the administration of polyI:C alone. LPS is known to be an agonist of TLR4, whereas polyI:C is an agonist of TLR3 and can signal through the cytoplasmic melanoma differentiation-associated protein 5 (MDA5) receptor [38,39]. Although the signaling pathways of TLR3, TLR4, and MDA5 activate similar sets of transcription factors (interferon regulatory factor 3, interferon regulatory factor 5, interferon regulatory factor 7, cAMP response element-binding protein, activating protein-1, nuclear factor kappa-light-chain-enhancer of activated B cells, and others), there is evidence that they regulate the expression of different sets of genes [38,40]. In the present paper, we hypothesized that low-dose LPS would enhance the inflammatory response to polyI:C. Some in vitro and in vivo studies have revealed a synergistic effect of TLR3 and TLR4 signaling cascades that results in cytokine overproduction [41,42,43]. In a recent report by Monguió-Tortajada et al. [44], priming of TLR3 by means of polyI:C was shown to enhance the inflammatory response of human monocytes to a subsequent challenge with low doses of LPS. At the same time, those authors noted that high production of TNF and IL6 in TLR3-primed cells in the presence of LPS is accompanied by their death via apoptosis, which can be reversed by blocking of tumor necrosis factor receptor (TNFR) I/II. Furthermore, a proinflammatory response of cells to the induction of various types of TLRs strongly depends on the sequence of treatments with various agents and the time interval between these treatments [41,45]. Accordingly, we can theorize that the simultaneous administration of polyI:C and LPS can either suppress TLR4 signaling owing to the presence of dominant cell activation through TLR3 or promote apoptosis.

Growing evidence suggests that TLR-mediated neuroinflammation in the enteric nervous system is an early event in cognitive decline, implying that neurocognitive diseases may have an origin outside the central nervous system. In the present article, we revealed a similarity of peripheral inflammation (a decrease in the number of leukocytes and lymphocytes and upregulation of genes *Tnf*, *Il-1b*, and *Il-6* in the spleen) between stimulation with a TLR3 agonist and stimulation with a TLR4 agonist in BTBR mice, but the neuroinflammation (activation of microglia and expression of genes *cFos*, *Tnf*, *Il-1b*, and *Il-6*) in brain tissues was found to depend only on TLR3 activation in BTBR mice. Besides, in BTBR mice, mainly in the hypothalamus but also in the frontal cortex, at 2 h after the combined administration of the two TLR mimetics, there was no upregulation of genes, and on the contrary, there was a tendency for attenuation of activation of proinflammatory pathways. This result points to a protective effect of LPS in the case of simultaneous administration with polyI:C. Most research articles compare the effects of activation of TLR4 and TLR3 on the development of neuropsychiatric diseases or evaluate the impact of these pathways on microglial cell lines in vitro. There is a concern that viral immortalization makes established microglial cell lines unrepresentative of their in vivo counterparts [46,47,48]. Quite often, the effects of two TLR mimetics are compared in the research on maternal or neonatal immune activation by means of behavioral outcomes in offspring [49,50]. Nonetheless, some brain cell markers are reported to be downregulated in LPS models, while no changes are seen in a polyI:C model. In other studies, identical outcomes in rats have been registered after LPS injection and after polyI:C injection. In general, LPS is widely used to characterize the effects of inflammation on various organ systems in offspring, whereas polyI:C is mostly applied to study neurodevelopmental disorders. Nevertheless, the two compounds share interactions with the neuroimmune pathway. A comparison of the effects of two mimetics in various animal strains having different sensitivity to TLR activation will help to determine whether the discrepancy in the results of administration of TLR agonists is due to a bias in the choice of an experimenter or to existing biological mechanisms underlying the development of various neuropsychiatric disorders.

Recently, scientists have come to a consensus that not only the adaptive immune system but also the innate one has properties of memory. This phenomenon is called training innate immunity [51]. Training innate immunity means that activation status of innate cells returns to baseline after the first stimulation prior to the second stimulation. By pathogen-associated molecular patterns such as those detected by TLRs, long-term functional reprogramming is triggered in transcriptional and metabolic pathways of leukocytes; the latter, when re-encountering the relevant pathogen, generate an enhanced immune response. Thus, persistent inflammation in neurodegenerative diseases may be linked with TLR activation earlier and persistent epigenetic changes in cells of the brain’s innate immune system, such as astrocytes and microglia. This is especially true for self-sustaining cell populations in organs, such as brain microglia. Consequently, the search for animal models that are sensitive to various types of TLR activation will clarify this phenomenon, identify a possible pathogenetic role of training innate immunity in the development of neuroinflammation in various neuropsychiatric diseases, and may help to find TLR antagonists that hold promise as immunotherapeutic agents for the treatment of neuroimmune inflammation. Thus, the BTBR strain can be used to investigate not only ASDs but also the effects of various drugs targeting TLR3-driven neuroinflammation.

Neuroinflammation, regardless of its etiology (sterile or infectious), can lead to neurodegenerative diseases, whose etiology is complicated and not yet fully understood. Astrocytes, microglia, oligodendrocytes, and glial cells express different types of TLRs [52] and serve as the brain’s local sentinel innate-immunity cells; the activation of these cells under diverse conditions gives rise to neuroinflammation. We found in this project that levels of expression of genes *Il-1β*, *Il-6, Tnf,* and *c-Fos* do not differ in the hypothalamus and PFC in a steady state between the two strains. Exposure to polyI:C and polyI:C together with a low dose of LPS, but not LPS alone, enhanced the expression of cytokine genes and *cFos* at 2 h after the injection, but their expression almost completely returned to baseline at 16 h after the injection. Of note, at 2 h after the injection, BTBR mice showed a less pronounced response in terms of *cFos* and *Tnf* expression in the hypothalamus and PFC and IL-6 expression in the PFC, but the overexpression of *Tnf* and *Il-6* (compared to the saline group) persisted for 16 h after the injection. These results may indicate both a slower immune response and a more prolonged inflammatory response in BTBR mice. Research on juvenile BTBR mice has also uncovered no changes in *Il-1β*, *Il-6*, and *Tnf* expression in a steady state in both the hypothalamus and cortex [25]; however, those authors registered upregulation of proinflammatory cytokine *Il-33* in both of these brain structures.

Our data showed that the levels of expression of *Aif1* and *Gfap* in BTBR mice is higher in a steady state in the hypothalamus, but in the PFC, *Aif1* expression is low. In response to polyI:C and polyI:C together with low-dose LPS, only in BTBR mice did the expression of *Aif1* go up in the hypothalamus, while in the PFC, it increased in mice of both strains. These results are indicative of region-specific reactivity of microglia in BTBR mice. Previously, quantitation of MHC-II expression using fluorescence intensity in cultured brain microglial cells showed that the fluorescence intensity is 35% higher, which indirectly indicates that microglia are more activated [25]. In our study, only BTBR mice showed an increase in *Gfap* expression at 2 h after injection of the mimetics. One of the functions of the intermediate filament network (GFAP, vimentin, nestin, and synemin) in astrocytes is to protect the cells from oxidative stress [53]; this stress is elevated in BTBR mice [54]. Activation of cytokines also raises the level of oxidative stress in the brain [55]. Accordingly, the glial activation in BTBR mice may be a protective response to an excessive enhancement of oxidative stress.

Furthermore, it is worth mentioning several considerations that may have influenced the findings as well as future directions of this research. First, regarding the choice of the mimetics’ doses, they vary widely in the literature on inflammation induction and can differ almost 10-fold both for polyI:C (2–12 µg/kg) and for LPS (100–1000 µg/kg) [35,56,57,58]. The dose that we chose for LPS (50 µg/kg) was low, and our dose of polyI:C (10 µg/kg) was moderate. The low dose of LPS and the moderate dose of polyI:C are consistent with induction of enough inflammation to replicate primarily a fatigue phenotype without the other unwanted side effects of a potent inflammatory change [56,57,58,59].

The other limitation of the interpretation of our data is the small number of time points for the analysis. Even though the period of 2–16 h covers the main effects of the mimetics on the activation of cytokine production, on the peripheral-blood leukocyte ratio, and on the peak of glia activation in the brain, the use of only two data points does not allow in some cases to correctly interpret interstrain differences.

Although in this work, hematological, immunological, and molecular biological approaches were utilized, an additional assessment of peripheral levels of circulating cytokines and chemokines, brain cytokines, and chemokines as well as examination of the morphology of microglia and astroglia by immunohistochemical analysis would increase translational value of the study. These approaches, in combination with a greater number of time points for measurements, will help to elucidate specific features of the immune system in BTBR mice.

## 4. Materials and Methods

### 4.1. Animals

In the experiment, male mice of strains BTBR T+Itpr3tf/J (BTBR) and C57BL/6J (B6) were used, who were kept under conditions of free access to water and feed at a multi-access center called the Center for Genetic Resources of Laboratory Animals, the Institute of Cytology and Genetics, the Siberian Branch of the Russian Academy of Sciences (ICG SB RAS) (RFMEFI62119 × 0023). Mice were housed in individually ventilated OptiMice cages (Animal Care Systems, Inc., Centennial, CO, USA) containing autoclaved dust-free birch bedding. The mice were 60 days old at the time of the experimental manipulations. Each mouse group included 4–8 BTBR or B6 males.

### 4.2. Experimental Design

BTBR mice received intraperitoneal injection of LPS (from *Escherichia coli* serotype O55:B5, 50 µg/kg), polyI:C (10 µg/kg), or their combination. The polyI:C dose was moderate in comparison to other studies [56,57,58], and the LPS dose was low in comparison to other studies (100–1000 µg/kg) [35], but as previously shown, this dose of LPS is sufficient to induce a moderate inflammatory response [59]. As a control, equivalent volumes of saline were administered. It is known that mammals with different genotypes show different responses of the immune system [60]. In this regard, B6 mice served as a control strain for comparing the levels of immune responses. At 2 and 16 h post-injection, the animals were killed by rapid decapitation (Figure 1). We chose these time intervals because activation of cytokine production is seen at 1.5–3.0 h after induction of inflammatory agents, then it decreases, but an elevated level of cytokines compared to baseline may persist for up to 24 h [29,36,56]; The peripheral-blood leukocyte ratio also changes after 2–3 h, and similar to cytokines, some changes can still be detected 24 h after induction [61]. The peak of glia activation and alterations in the profile of peripheral-blood lymphocytes are observed after 6–24 h [35,36,61]. All sampling procedures were performed in a cold room. Samples of trunk blood were collected from the mice; samples of the spleen, hypothalamus, and prefrontal cortex (PFC) were excised as described previously [62], then they were immediately frozen in liquid nitrogen and stored at −80 °C until analysis.

### 4.3. Complete Blood Count

After the animals were killed, blood was collected into 1.5 mL test tubes (Eppendorf, Germany) containing 100 µL of 0.5 M EDTA. The complete blood count was performed on a BC-2800 Vet (Mindray auto hematology analyzer, Mahwah, NJ, USA) at the multi-access center SPF Vivarium of the ICG SB RAS. For the analysis of peripheral inflammation, absolute numbers of lymphocytes, granulocytes, monocytes, and platelets were determined, expressed in ×10^9^ cells/L.

### 4.4. Flow-Cytometric Analysis

After the euthanasia, 200 µL of blood was collected into 1.5 mL tubes (Eppendorf, Germany) containing 100 µL of 0.5 M EDTA. For flow cytometry, erythrocytes were disrupted using RBC Lysing Buffer (0.15 M ammonium chloride, 10 mM sodium bicarbonate, and 0.1 mM EDTA). After the lysis of erythrocytes, the remaining cells were washed twice.

Profiling of the peripheral blood lymphocyte population was performed on a BD FACSCanto™ II flow cytometer. Staining was performed according to the manufacturer’s protocol with an appropriate combination of fluorescent monoclonal antibodies: anti-mouse CD45-PE/Cy7, anti-mouse CD19-FITC, anti-mouse CD4, anti-mouse CD8-PE/Cy7, and anti-mouse CD3-PE (all from BioLegend, San Diego, CA, USA).

### 4.5. Analysis of Gene Expression

Isolation of total RNA was performed using PureZol (Bio-Rad, Hercules, CA, USA) according to the manufacturer’s protocol. The resulting RNA was treated with DNase I, RNase-free (Thermo Fisher Scientific, Waltham, MA, USA) and purified on RNA Clean beads (Beckman Coulter, Germany). To determine the concentration of RNA and its purity (traces of proteins), a NanoDrop 2000 spectrophotometer was utilized (Thermo Fisher Scientific, Waltham, MA, USA).

For a reverse-transcription reaction, 1.0 µL of 100 µM hexanucleotide primers was added to 1 µg of RNA and diluted with double-distilled water to a total volume of 13 µL. The resultant mixture was incubated at 65 °C for 5 min, then kept on ice for 5 min. Next, 4 µL of 5× reaction buffer, 2.0 µL of 10 mM dNTP Mix, 1.0 µL of Revert Aid enzyme (200 U/µL), and double-distilled water were added to the samples (to attain total volume 20 µL). This solution was mixed and incubated at 25 °C for 5 min, then at 42 °C for 60 min, and finally for 10 min at 70 °C. After this reaction, all samples were diluted 5-fold, and real-time PCR was carried out.

The main mediators of neuroinflammation are cytokines interleukin-1β (IL-1β), tumor necrosis factor (TNF), and IL-6. Their main sources in the central nervous system are astrocytes and microglia. Therefore, for quantitative analysis of expression levels, the following genes were chosen: *Gfap*, a marker of reactive astrocytes; *Aif1*, a marker of activated microglia; *cFos*, a marker of neuronal activation; and proinflammatory-cytokine genes *Tnf*, *Il-1β*, and *Il-6* (Table 2). The real-time PCR included TaqMan fluorescent probes. The gene expression was assessed relative to mRNA levels of housekeeping genes (*Hk1*, *Pik*, and *Rab*). For the PCR, the BioMaster HS-qPCR kit (2×) was employed on a Real-Time CFX96 Touch thermal cycler (Bio-Rad Laboratories, Hercules, CA, USA) according to the protocol 95 °C-15″, 60 °C-20″. For each cDNA sample, all the analyses were performed in triplicate.

### 4.6. Statistical Analysis

The data were tested for normality of distribution by the Kolmogorov–Smirnov test. Statistical evaluation of the obtained data was carried out in the STATISTICA software (ver.8.0; StatSoft, Inc., 2007, Tulsa, OK, USA) by two-way ANOVA (the group and strain of animals served as the factors), and the significance of differences between groups was assessed by Fisher’s least significant difference (LSD) test.

## 5. Conclusions

BTBR mice have several specific features in terms of the development of mimetic-induced peripheral inflammation and of inflammation in the brain. BTBR mice may be a promising model for testing neuroinflammation or the effectiveness of vaccines. As far as we know the immune features of BTBR mice, it is also necessary to study specific features of the formation of adaptive immunity. Taken together, these data will improve the understanding of the translational value of the BTBR strain.

## Figures and Tables

**Figure 1 ijms-23-15577-f001:**
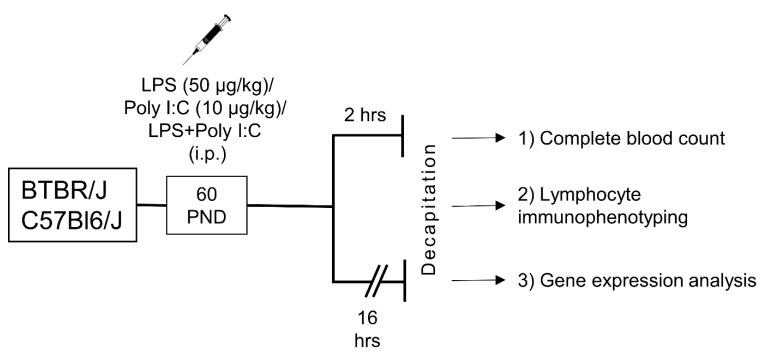
Experimental design. Adult male mice of both strains were injected with lipopolysaccharide (LPS), polyinosinic:polycytidylic acid (Poly I:C), or a combination thereof. After 2 or 16 h, blood and tissues were sampled for assays. PND-postnatal day.

**Figure 2 ijms-23-15577-f002:**
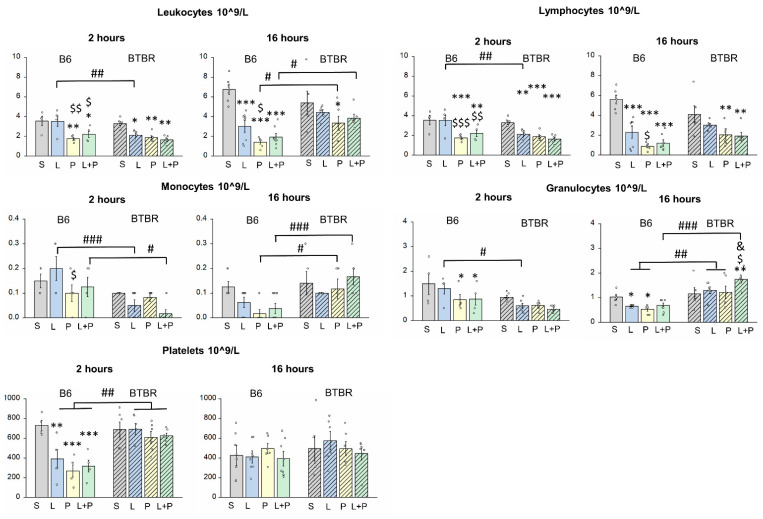
Numbers of leukocytes, lymphocytes, monocytes, granulocytes, and platelets in the blood of B6 and BTBR mice following systemic injection of polyI:C (P), LPS (L), or their combination (L+P). The control group is marked as S (received saline). The induced inflammation altered the counts of blood cells in different directions depending on the strain and time after injection. Factorial ANOVA, mean ± SEM, *n* = 4–8 per group. * *p* < 0.05, ** *p* < 0.01, *** *p* < 0.001 in comparison with saline control; $ *p* < 0.05, $$ *p* < 0.01, $$$ *p* < 0.001 as compared to the LPS group; & *p* < 0.05 as compared to the polyI:C group; # *p* < 0.05, ## *p* < 0.01, ### *p* < 0.001 in comparison with the corresponding B6 group according to Fisher’s LSD test as a post hoc analysis.

**Figure 3 ijms-23-15577-f003:**
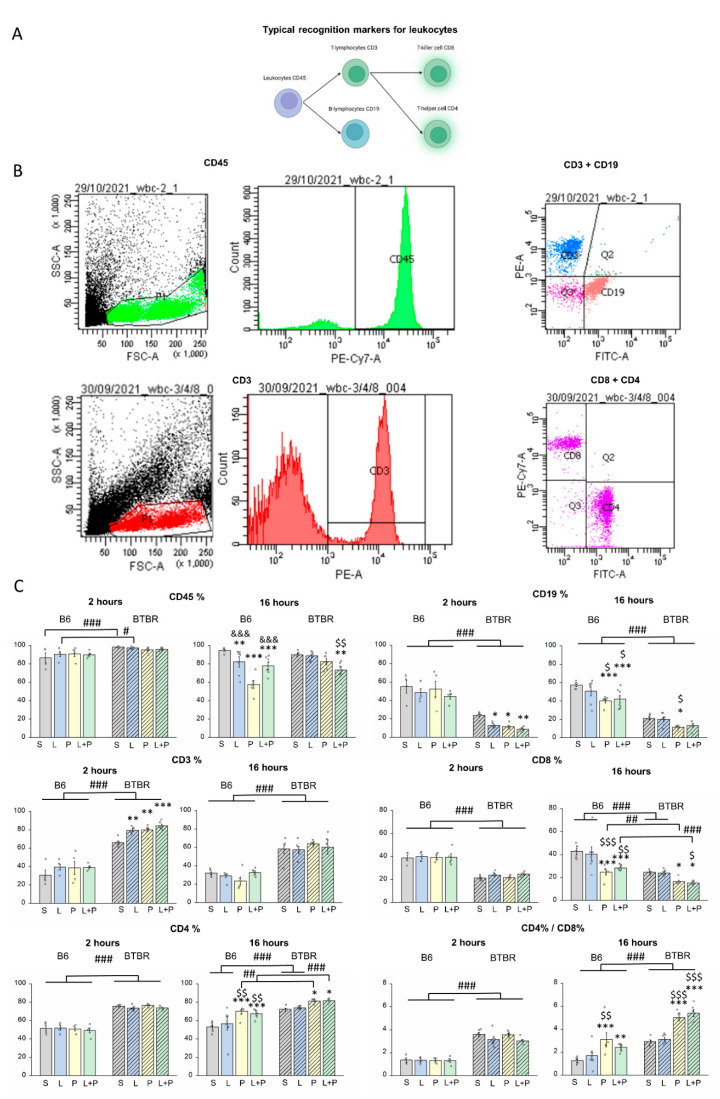
(**A**) Results of different types of lymphocytes. Typical markers of lymphocytes were assayed; (**B**) the gating strategy for evaluating the distribution CD3^+^ T cells, CD19^+^ B cells, CD4^+^ T helper cells, and CD8^+^ cytotoxic T lymphocytes; and (**C**) changes in the ratio of types of lymphocytes after the induction of inflammation by administration of polyI:C (P), LPS (L), or their combination (L+P). The control group is marked as S (received saline). Factorial ANOVA, mean ± SEM, *n* = 4–8 per group. * *p* < 0.05, ** *p* < 0.01, *** *p* < 0.001 in comparison with saline control; $ *p* < 0.05, $$ *p* < 0.01, $$$ *p* < 0.001 as compared to the LPS group; &&& *p* < 0.001 as compared to the polyI:C group; # *p* < 0.05, ## *p* < 0.01, ### *p* < 0.001 in comparison with the corresponding B6 group according to Fisher’s LSD test as a post hoc analysis.

**Figure 4 ijms-23-15577-f004:**
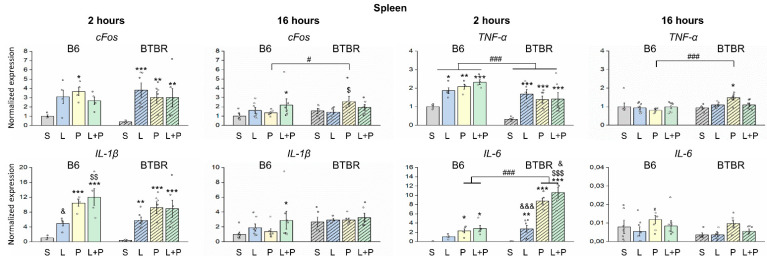
Changes in splenic levels of neuronal early response gene (*cFos)* and cytokines’ mRNAs after the induction of inflammation via administration of polyI:C (P), LPS (L), or their combination (L+P). The control group is marked as S (received saline). The mRNAs were quantified by RT-PCR. Factorial ANOVA, mean ± SEM, *n* = 4–8 per group. * *p* < 0.05, ** *p* < 0.01, *** *p* < 0.001 as compared to saline control; $ *p* < 0.05, $$ *p* < 0.01, $$$ *p* < 0.001 compared to the LPS group; & *p* < 0.05, &&& *p* < 0.001 as compared to the polyI:C group; # *p* < 0.05, ### *p* < 0.001 in comparison with the B6 strain according to Fisher’s LSD test as a *post hoc* analysis.

**Figure 5 ijms-23-15577-f005:**
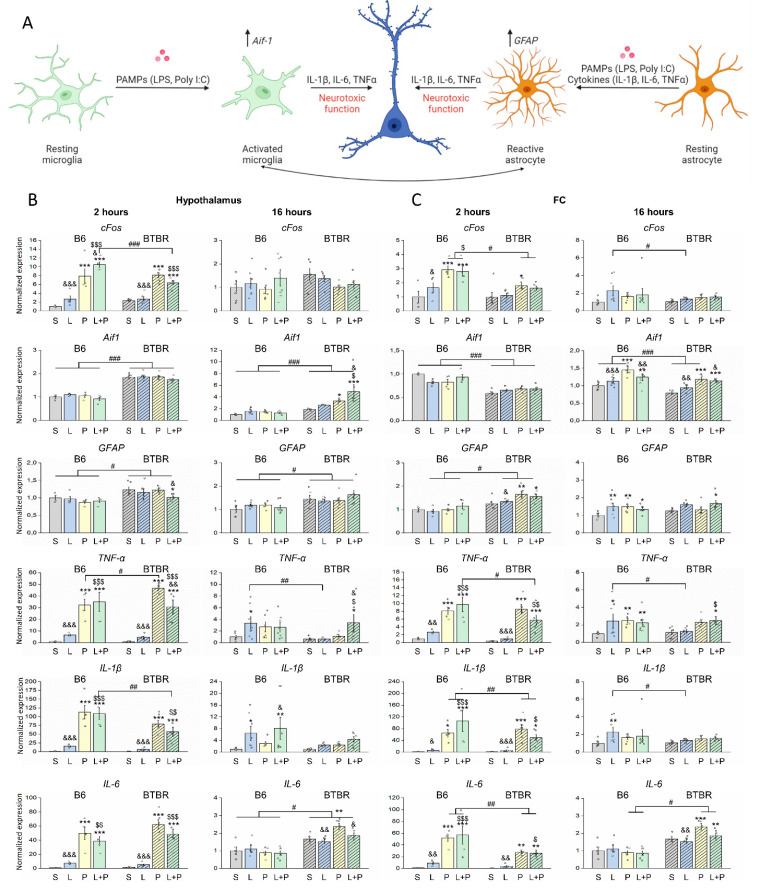
(**A**) The glial reaction to the acute inflammation; and (**B**,**C**) regional neuronal, microglial, and astrocyte activation markers and cytokines in the mouse brain after systemic injection of polyI:C (P), LPS (L), or their combination (L+P). The control group is marked as S (received saline). The mRNAs were quantitated by RT-PCR in the hypothalamus and prefrontal cortex (PFC) of B6 and BTBR mice. polyI:C and its combination with LPS significantly raised levels of *cFos,* allograft inflammatory factor 1 (*Aif1*), glial fibrillary acidic protein Gfap, tumor necrosis factor (*Tnf*), interleukin-1 beta (*Il-1β*), and interleukin-6 (*Il-6*), mRNAs. Factorial ANOVA, mean ± SEM, *n* = 4–8 per group. * *p* < 0.05, ** *p* < 0.01, *** *p* < 0.001 as compared to saline control; $ *p* < 0.05, $$ *p* < 0.01, $$$ *p* < 0.001 in comparison with the LPS group; & *p* < 0.05, && *p* < 0.01, &&& *p* < 0.001 as compared to the polyI:C group; # *p* < 0.05, ## *p* < 0.01, ### *p* < 0.001 as compared to the corresponding B6 group according to Fisher’s LSD test as a *post hoc* analysis.

**Table 1 ijms-23-15577-t001:** Results of statistical analysis of the complete blood count and data on different types of lymphocytes.

Parameters	Time after Injection	Strain	Group	Strain × Group
Leukocytes, ×10^9^/L	2 h	F(1,39) = 10.015 ***p *****< 0.01**	F(1,39) = 9.125 ***p *****< 0.001**	F(3,36) = 1.650 *p* > 0.5
	16 h	F(1,45) = 6.571 ***p *****< 0.05**	F(1,45) = 17.703 ***p *****< 0.001**	F(3,42) = 4.102 ***p***** < 0.05**
Lymphocytes, ×10^9^/L	2 h	F(1,39) = 6.092 ***p***** < 0.05**	F(1,39) = 12.675 ***p***** < 0.001**	F(3,36) = 2.267 *p* > 0.5
	16 h	F(1,45) = 0.813 *p* > 0.5	F(1,45) = 26.162 ***p***** < 0.001**	F(3,42) = 3.746 ***p***** < 0.05**
Monocytes, ×10^9^/L	2 h	F(1,39) = 15.700 ***p***** < 0.001**	F(1,39) = 1.689 *p* > 0.5	F(3,36) = 2.102 *p* > 0.5
	16 h	F(1,45) = 14.782 ***p***** < 0.001**	F(1,45) = 2.329 *p* > 0.5	F(3,42) = 2.130 *p* > 0.5
Granulocytes, ×10^9^/L	2 h	F(1,39) = 11.462 ***p***** < 0.01**	F(1,39) = 3.217 ***p***** < 0.05**	F(3,36) = 0.508 *p* > 0.5
	16 h	F(1,45) = 42.792 ***p***** < 0.001**	F(1,45) = 2.357 *p* > 0.5	F(3,42) = 3.916 ***p***** < 0.05**
Platelets, ×10^9^/L	2 h	F(1,39) = 22.074 ***p***** < 0.001**	F(1,39] = 6.260 ***p***** < 0.01**	F(3,36) = 3.484 ***p***** < 0.05**
	16 h	F(1,45) = 1.767 *p* > 0.5	F(1,45) = 0.432 *p* > 0.5	F(3,42) = 0.408 *p* > 0.5
CD45^+^ %	2 h	F(1,39) = 20.357 ***p***** < 0.001**	F(1,39) = 0.185 *p* > 0.5	F(3,36) = 1.134 *p* > 0.5
	16 h	F(1,43) = 5.077 ***p***** < 0.05**	F(1,43) = 16.141 ***p***** < 0.001**	F(3,40) = 7.920 ***p***** < 0.001**
CD3^+^ %	2 h	F(1,39) = 263.534 ***p***** < 0.001**	F(1,39) = 5.700 ***p***** < 0.01**	F(3,36) = 0.681 *p* > 0.5
	16 h	F(1,43) = 141.102 ***p***** < 0.001**	F(1,43) = 0.323 *p* > 0.5	F(3,40) = 1.471 *p* > 0.5
CD19^+^ %	2 h	F(1,39) = 155.891 ***p***** < 0.001**	F(1,39) = 3.597 ***p***** < 0.05**	F(3,36) = 0.436 *p* > 0.5
	16 h	F(1,43) = 223.312 ***p***** < 0.001**	F(1,43) = 9.135 ***p***** < 0.001**	F(3,40) = 0.648 *p* > 0.05
CD8^+^ %	2 h	F(1,39) = 172.804 ***p***** < 0.001**	F(1,39) = 0.626 *p* > 0.5	F(3,36) = 0.266 *p* > 0.5
	16 h	F(1,43) = 53.095 ***p***** < 0.001**	F(1,43) = 12.821 ***p***** < 0.001**	F(3,40) = 1.267 *p* > 0.05
CD4^+^ %	2 h	F(1,39) = 375.799 ***p***** < 0.001**	F(1,39) = 0.669 *p* > 0.5	F(3,36)= 0.483 *p* > 0.5
	16 h	F(1,43) = 55.375 ***p***** < 0.001**	F(1,43) = 10.173 ***p***** < 0.001**	F(3,40) = 0.645 *p* > 0.05
CD8^+^ %/CD4^+^ %	2 h	F(1,39) = 130.405 ***p***** < 0.001**	F(1,39) = 0.324 *p* > 0.5	F(3,36) = 0.073 *p* > 0.5
	16 h	F(1,43) = 12.385 ***p***** < 0.01**	F(1,43) = 3.020 ***p***** < 0.05**	F(3,40) = 1.021 *p* > 0.05

**Table 2 ijms-23-15577-t002:** The list of primers for PCR.

Gene	Sequence 5′→3′	
*Pik3c3*	Probe	HEX-ACTTGATGGTTGAGTTTCGCTGTGT-BHQ1
	For	GGATTGGCTGGACAGATT
	Rev	CTCCTTGTCATCGCACTT
*Hk1*	Probe	Cy5-CCTTCTCGTTTCCCTGCAAG-BHQ2
	For	ACATTGTCTCCTGCATCTCC
	Rev	GCTTTGAATCCCTTTGTCCAC
*Rab22a*	Probe	Cy5-AGCATCGTGTGGCGGTTTGTG-BHQ2
	For	GATACGGGTGTGGGTAAATC
	Rev	CTGGACAGTCTTGGTCATAAA
*cFos*	Probe	ROX-CGTCATCCTCCCGCTGCA-BHQ2
	For	CGGGTTTCAACGCCGACTA
	Rev	TTGGCACTAGAGACGGACAGA
*Aif1*	Probe	ROX-AGAGAGGCTGGAGGGGATC-BHQ2
	For	GCTTTTGGACTGCTGAAGGC
	Rev	GAAGGCTTCAAGTTTGGACG
*Gfap*	Probe	ROX-GCAAGAGACAGAGGAGTGG-(BHQ-2)
	For	CCTGAGAGAGATTCGCACTC
	Rev	GACTCCAGATCGCAGGTCAAG
*TNF*	Probe	ROX-CGAGTGACAAGCCTGTAGC-BHQ2
	For	CATCAGTTCTATGGCCCAGACCCT
	Rev	GCTCCTCCACTTGGTGGTTTGCTA
*IL-1β*	Probe	ROX-CTGCTTCCAAACCTTTGACCTGG-BHQ2
	For	CCTGTTCTTTGAAGTTGACGG
	Rev	CTGAAGCTCTTGTTGATGTGC
*IL-6*	Probe	ROX-CTGGGAAATCGTGGAAATGAG-BHQ2
	For	CAGACCTGTCTATACCACTTCAC
	Rev	GGTACTCCAGAAGACCAGAGG

## Data Availability

Data supporting the findings of this study are available upon reasonable request.

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
