# Peer review of "Unique Features of the Immune Response in BTBR Mice"

_ijms, 2022, doi:10.3390/ijms232415577_

Round 1

Reviewer 1 Report

The manuscript by Mutovina and co-workers addresses an interesting and pertinent topic, since it is relevant to study the interconnection between immune dysfunction, neuroinflammation and behavioral changes, in order to unveil the molecular basis of neurocognitive disorders and psychopathologies.

This manuscript reports a well-reasoned animal study, combining hematological, immunological and molecular biology approaches. The results have potential applicability in the clinical settings, for which it represents an added value. The draft is well structured.

Despite its current flaws, which I detail below, I believe that this paper fits the scope of the IJMS and that it could be considered for publication, once the following concerns are met.

General remarks

·      I recommend that the manuscript is revised for formatting, since several aspects, including the font and text alignment, do not comply with those of the IJMS template.

·      According to the instructions for authors, acronyms/abbreviations/initialisms should be defined the first time they appear in each of three sections: the abstract; the main text; the first figure or table. When defined for the first time, the acronym/abbreviation/initialism should be added in parentheses after the written-out form. Please make sure this is done for all acronyms/abbreviations/initialisms (e.g., MDA5, IRF, CREB, etc.).

Introduction

·      Please provide a rationale for LPS and Poly I:C dose choice, as well as for the selection of the post-injection timepoints assayed.

·      Figure 1: please specify the meaning of “60 PND” (60 postnatal days, I suppose) in the legend of the figure.

·      Please provide a rationale for the selection of brain structures assayed (hypothalamus and prefrontal cortex).

Discussion

·      Please acknowledge study limitations and future prospects/opportunities for complementary assays.

Materials and Methods

·      Section 4.1: how was the total number of animals in the study defined? Please add some remarks on sample size.

·      Line 434: please italicize “Escherichia coli”.

·      Line 435: please correct LPS dose units.

·      Line 444: please remove the “A” from the title of section 4.3.

Conclusions

·      This section is not supposed to provide contents that were not previously approached within the manuscript, such as the specificities of the genomic differences between both mice strains. Please integrate this kind of information into the “Discussion” section, while leaving a mere summary of the main conclusions for this one.

References:

·      The number of each reference appears twice.

Author Response

Dear Editor and Referees:

Thank you for giving us the opportunity to submit a revised draft of our manuscript titled “Unique features of the immune response in BTBR mice.” We are thankful to you for reviewing the manuscript and sharing your valuable comments and concerns with us. We have been able to incorporate most of the suggestions provided by the reviewers. We have highlighted the revisions within the manuscript.

Below, highlighted in red, are point-by-point responses to the reviewers’ comments and concerns.

Sincerely,

Vasiliy Reshetnikov

The manuscript by Mutovina and co-workers addresses an interesting and pertinent topic, since it is relevant to study the interconnection between immune dysfunction, neuroinflammation and behavioral changes, in order to unveil the molecular basis of neurocognitive disorders and psychopathologies.

This manuscript reports a well-reasoned animal study, combining hematological, immunological and molecular biology approaches. The results have potential applicability in the clinical settings, for which it represents an added value. The draft is well structured.

Despite its current flaws, which I detail below, I believe that this paper fits the scope of the IJMS and that it could be considered for publication, once the following concerns are met.

General remarks

  • I recommend that the manuscript is revised for formatting, since several aspects, including the font and text alignment, do not comply with those of the IJMS template.

Reply: Thank you for noticing. The formatting of the manuscript has been carefully checked and brought to the MDPI standards.

  • According to the instructions for authors, acronyms/abbreviations/initialisms should be defined the first time they appear in each of three sections: the abstract; the main text; the first figure or table. When defined for the first time, the acronym/abbreviation/initialism should be added in parentheses after the written-out form. Please make sure this is done for all acronyms/abbreviations/initialisms (e.g., MDA5, IRF, CREB, etc.).

Reply: The text has been corrected.

 Introduction

  • Please provide a rationale for LPS and Poly I:C dose choice, as well as for the selection of the post-injection timepoints assayed.

Reply: Thank you for pointing this out. We now justify the choice of doses and post-injection timepoints in subsection 4.2 “Experimental design”.

      Figure 1: please specify the meaning of “60 PND” (60 postnatal days, I suppose) in the legend of the figure.

Reply: The legend of Figure 1 has been revised.

  • Please provide a rationale for the selection of brain structures assayed (hypothalamus and prefrontal cortex).

Reply: According to this comment, we have added more information in section 1 “Introduction”.

Discussion

  • Please acknowledge study limitations and future prospects/opportunities for complementary assays.

 Reply: In the Discussion section, we have inserted some limitations and prospects.

Materials and Methods

  • Section 4.1: how was the total number of animals in the study defined? Please add some remarks on sample size.
  • Line 434: please italicize “Escherichia coli”.
  • Line 435: please correct LPS dose units.
  • Line 444: please remove the “A” from the title of section 4.3.

Reply: The text has been corrected.

 Conclusions

  • This section is not supposed to provide contents that were not previously approached within the manuscript, such as the specificities of the genomic differences between both mice strains. Please integrate this kind of information into the “Discussion” section, while leaving a mere summary of the main conclusions for this one.

Reply: We agree with the reviewer’s remark. The conclusions have been updated.

As far as we know the immune features of BTBR mice, it is also necessary to study specific features of the formation of adaptive immunity. Taken together, these data will improve the understanding of translational value of the BTBR strain.

References:

  • The number of each reference appears twice.

Reply: The duplicates of reference numbers have been corrected.

Reviewer 2 Report

This is an interesting study that compare immune responses to two different inflammation inducers. Comparisons have been made on two different BTBR and C57BL6/J mouse lines. The results are presented well and is well discussed. This study makes a good advancement towards understanding immune response in different mouse lines.  

Author Response

Dear  Referee:

We are thankful to you for reviewing the manuscript titled “Unique features of the immune response in BTBR mice.” and sharing your valuable comments and concerns with us.